# Predicting the Efficacy of SBRT for Lung Cancer with ^18^F-FDG PET/CT Radiogenomics

**DOI:** 10.3390/life13040884

**Published:** 2023-03-27

**Authors:** Kuifei Chen, Liqiao Hou, Meng Chen, Shuling Li, Yangyang Shi, William Y. Raynor, Haihua Yang

**Affiliations:** 1Taizhou Hospital of Zhejiang Province, Shaoxing University, Taizhou 317000, China; 2Key Laboratory of Radiation Oncology of Taizhou, Radiation Oncology Institute of Enze Medical Health Academy, Department of Radiation Oncology, Taizhou Hospital Affiliated to Wenzhou Medical University, Taizhou 317000, China; 3Department of Radiation Oncology, University of Arizona, Tucson, AZ 85724, USA; 4Department of Radiology, Rutgers Robert Wood Johnson Medical School, New Brunswick, NJ 08901, USA

**Keywords:** ^18^F-fluorodeoxyglucose positron emission tomography/computed tomography, radiogenomics, stereotactic body radiation therapy, epidermal growth factor receptor, lung cancer

## Abstract

Purpose: to develop a radiogenomic model on the basis of ^18^F-FDG PET/CT radiomics and clinical-parameter EGFR for predicting PFS stratification in lung-cancer patients after SBRT treatment. Methods: A total of 123 patients with lung cancer who had undergone ^18^F-FDG PET/CT examination before SBRT from September 2014 to December 2021 were retrospectively analyzed. All patients’ PET/CT images were manually segmented, and the radiomic features were extracted. LASSO regression was used to select radiomic features. Logistic regression analysis was used to screen clinical features to establish the clinical EGFR model, and a radiogenomic model was constructed by combining radiomics and clinical EGFR. We used the receiver operating characteristic curve and calibration curve to assess the efficacy of the models. The decision curve and influence curve analysis were used to evaluate the clinical value of the models. The bootstrap method was used to validate the radiogenomic model, and the mean AUC was calculated to assess the model. Results: A total of 2042 radiomics features were extracted. Five radiomic features were related to the PFS stratification of lung-cancer patients with SBRT. T-stage and overall stages (TNM) were independent factors for predicting PFS stratification. AUCs under the ROC curve of the radiomics, clinical EGFR, and radiogenomic models were 0.84, 0.67, and 0.86, respectively. The calibration curve shows that the predicted value of the radiogenomic model was in good agreement with the actual value. The decision and influence curve showed that the model had high clinical application values. After Bootstrap validation, the mean AUC of the radiogenomic model was 0.850(95%CI 0.849–0.851). Conclusions: The radiogenomic model based on ^18^F-FDG PET/CT radiomics and clinical EGFR has good application value in predicting the PFS stratification of lung-cancer patients after SBRT treatment.

## 1. Introduction

Globally, lung cancer is the most common cancer by incidence and the leading cause of cancer-related deaths, with more than 2.2 million new cases and 1.8 million deaths yearly [1]. Stereotactic body radiation therapy (SBRT) demonstrated excellent rates of local control (LC), progress-free survival (PFS), and overall survival (OS), compared with those obtained with surgery or standard radiotherapy in multiple prospective trials [2,3,4,5]. SBRT is a safe and local alternative treatment option for patients with nonsmall-cell lung cancer (NSCLC) who are medically inoperable or reluctant to undergo surgery and pulmonary oligometastasis [6,7]. Therapeutic outcomes still greatly vary among these patients. A new prediction model is needed to indicate responsiveness to SBRT, although the peripheral-blood model could solidly predict the efficacy of SBRT in lung cancer [8].

Recently, targeted therapy has been used relatively often in treating individuals afflicted with lung adenocarcinoma [9]. In many patients with gene variants, targeted therapy with tyrosine kinase inhibitors (TKIs) may significantly improve survival and quality of life [10]. According to retrospective findings, patients with advanced epidermal growth factor receptor (EGFR) mutant NSCLC may benefit from combining SBRT with ongoing TKI therapy, which prolongs PFS and overall survival [11,12,13].

^18^F-fluorodeoxyglucose positron emission tomography/computed tomography (^18^F-FDG PET/CT) is used in cancer diagnosis and treatment monitoring, and has had widespread use as a technique for staging, restaging, and response assessment in lung cancer [14]. Radiomics uses features identified from imaging data as biomarkers to identify correlations with pathological or molecular reference, treatment response, or survival outcomes [15,16]. Current findings show that radiomics may effectively grade tumors, evaluate adverse treatment-related effects, and forecast the clinical endpoints of patients with lung cancer [17,18,19]. 

Radiogenomics refers to the combination of imaging-derived features ith genetic data to identify clinically meaningful correlations. This method has the benefit of acquiring information about the whole tumor that may be used to track the progress of therapy [20]. This study aims to develop and validate an ^18^F-FDG PET/CT radiogenomic model on the basis of ^18^F-FDG PET/CT radiomics and clinical-parameter EGFR to predict the PFS stratification of SBRT in lung-cancer patients.

## 2. Materials and Methods

### 2.1. Patients and Inclusion Criteria

A total of 475 individuals underwent SBRT from September 2014 to December 2021. Inclusion criteria: (1) pathologically confirmed lung cancer and complete EGFR results before SBRT; (2) available ^18^F-FDG PET/CT images; (3) SBRT within 1 month after the completion of ^18^F-FDG PET/CT; (4) complete clinical and imaging data. Exclusion criteria: (1) poor ^18^F-FDG PET/CT picture quality (prominent artifacts), hindering texture analysis and tumor region of interest (ROI) delineation, and preventing imaging radiomics analysis; (2) more than 1 month between ^18^F-FDG PET/CT imaging and SBRT. A total of 123 patients participated in the study (Figure 1). Disease staging was performed according to the 8th edition of the American Joint Committee on Cancer Staging (AJCC) Manual [21]. Time until disease progression (i.e., the growth of a residual tumor or the formation of a new metastatic lesion), death, or censoring at the date of the final follow-up after SBRT had been initiated was used to calculate PFS.

### 2.2. Imaging Protocol for ^18^F-FDG PET/CT

All patients underwent ^18^F-FDG PET/CT according to the European Association of Nuclear Medicine guidelines [22]. Under the ^18^F-FDG PET/CT protocol, each patient in the study was scanned using the scanner manufactured by American General Electric Discovery Elite at Taizhou Hospital. Shanghai Sinovac Pharmaceutical Co., Ltd. was responsible for producing and distributing ^18^F-FDG. Before receiving an injection of ^18^F-FDG (0.1 mCi/kg), all patients were required to fast for at least 4 h before the assessment and to maintain their blood glucose levels below 10 millimoles per liter. One hour after the injection, PET/CT scans were taken from the base of the head to the middle of the femur. CT scans (scan parameters: slice thickness, 3.75 mm; pitch, 0.984; tube voltage, 120 kV; noise value (NI), 12.82; tube current, 70–180 mA) were performed after the examination range had been determined; scanning time, 20–30s. After that, PET images of the patient’s whole body were obtained using 8–10 beds in three-dimensional mode, a rate of 1.5 min per bed position. The inspection was completed in 1.5 h, and the measurement was carried out in 1.5 h. The CT scans were used to calculate the attenuation correction for the PET images. To produce fusion pictures in three planes (transverse, sagittal, and coronal), PET/CT scans were reconstructed using an iterative technique.

### 2.3. Stereotactic Body Radiation Therapy (SBRT)

The patients lay on their backs with their arms propped over their heads. A full-body mask composed of thermoplastic rendered them immobile. Four-dimensional computed tomography (4D CT) images with motion management were collected in accessible quiet breathing mode. In the simulation procedure, an unenhanced CT (Discovery CT590 RT, GE) scan was taken for the dosage estimations with a slice thickness of 5 mm. CT windows of the lungs were used to create the tumor’s gross volume. The internal tumor volume (ITV) was mapped out using the 4D CT scan’s maximal intensity projection phase and fused with unenhanced CT images. A 5 mm margin was added to the ITV to reduce the setup error and then obtain the PTV. The organ at risk (OAR) was delineated on the unenhanced CT images by senior radiation oncologists, including bilateral lungs, esophagus, heart, and spinal cord. Every SBRT treatment was administered using 9–11 fields of fixed coplanar static-intensity-modulated radiation treatment. Pinnacle v. 9.10 (Philips Medical Systems, Milpitas, CA, USA) was used to calculate dosing using the collapsed cone convolutional method (Philips Medical Systems, Milpitas, CA, USA). Physicians determine the dose scheme of patients with SBRT according to the recommendations of the National Comprehensive Cancer Network (NCCN) guidelines. The size and location of the target lesions, the age of patients, cardiopulmonary function, and other factors must be considered in clinical treatment decisions. This study’s prescription doses for central and peripheral lung cancer were 50–70 Gy in 5–10 fractions. A radiation oncologist evaluated the treatment plan following RTOG 0915. Before treatment, the dosimetry parameters were examined to ensure meeting the protocol requirements. Cone-beam CT was used for imaging guidance throughout each treatment to guarantee that the isocenter of each patient’s treatment was precisely aligned.

### 2.4. Segmentation, and Feature Extraction and Selection

#### 2.4.1. Segmentation

The radiomic workflow comprises image acquisition, ROI segmentation, feature extraction and selection, and model development and validation [23,24] (Figure 2). Without clinical and pathologic information, radiation oncologists (with more than 10 years of experience) segmented the ROIs using ITK-SNAP software (version 3.8.0-beta; http://www.itksnap.org, accessed on 12 April 2022) [25]. 

#### 2.4.2. Feature Extraction

Before extracting the features, images were resampled to isometric voxels of 1 × 1 × 1 mm^3^, and Hounsfield units were binarized to discrete values of 25 HU. Radiomic features were extracted by using the Pyradiomics package (version 3.0.1) for Python (version 3.7.6) [26] following the standards set out by the Image Biomarker Standardization Initiative (IBSI) [27]. Radiomic features include intensity-statistics, shape, and texture features. Texture features consist of a gray-level cooccurrence matrix (GLCM), gray-level dependency matrix (GLDM), gray-level run length matrix (GLRLM), and gray-level size zone matrix (GLSZM). In addition, all categories of radiomic features except for shape features were extracted from three different image types: original, Laplacian of the Gaussian (log; sigma parameters were set to 3, 5, 10), and wavelet images. As a result, 2042 radiomic features were extracted from each patient’s PET/CT image. Detailed information on radiomic features can be found on the official website (https://pyradiomics.readthedocs.io/en/latest/index.html, accessed on 13 July 2022).

#### 2.4.3. Feature Selection and Model Construction

The redundancy analysis of radiomic features was performed, normal distribution was analyzed by the Pearson correlation coefficient, and non-normal distribution was analyzed via the Spearman correlation coefficient. Any features with higher average absolute correlation coefficients than 0.9 were considered redundant and excluded. Then, the least absolute shrink and selection operator (LASSO) was used to select the most essential features. This obtains a more refined model by constructing a penalty function that compresses some regression coefficients, i.e., forces the sum of the absolute values of the coefficients to be less than some fixed value while setting some regression coefficients to zero. Thus, it retains the advantage of subset shrinkage and is a biased estimator dealing with data with complex covariance. Then, the screened significant radiomic features were applied to set the radiomic model. The clinical data were collected from electronic patient records, including age, sex, Karnofsky performance status (KPS), Eastern Cooperative Oncology Group (ECOG), T classification, overall stage (TNM), tumor location, pathological type, smoking history, biological equivalent dose (BED), maximal standardized uptake value (SUVmax), and EGFR mutation. The logistic regression analysis of the clinical parameters was performed to find independent predictors of stratified PFS to construct a clinical EGFR model. In addition, we set a combined model according to clinical features, EGFR, and radiomic features.

### 2.5. Model Evaluation and Validation 

To ascertain the models’ clinical value, we produced the clinical impact curve (CIC) of and performed decision curve analysis (DCA) on the models, evaluating the net benefits. The receiver operating characteristic (ROC) curve analysis, the area under the curve (AUC) value, and sensitivity, specificity, accuracy, positive predictive value (PPV), and negative predictive value (NPV) were used to evaluate the predictive performance of the model. 

The bootstrap method for internal validation constructs a bootstrap resampling sample with the same sample size in the model development cohort. This sample was used as the validation set. The model development cohort was used as the training set to evaluate the model performance and repeat this process n times to obtain the performance of the model in internal validation. The bootstrap method was used to validate the radiogenomic model in this study [28]. Internal validation was performed with bootstrapping (500 samples). The AUCs of the radiogenomic and the other two models were compared using the DeLong test.

### 2.6. Statistical Analysis

All statistical analyses were performed in SPSS software (Version 23.0), R software (version 4.0.1), and Python (version 3.7.6). Differences in the categorical variables were evaluated using the chi-squared test comparing long and short PFS stratification. DeLong tests were used to compare the AUC values of the two models and see if there was any statistically significant difference. The level of statistical significance was defined as a *p*-value less than 0.05 on the basis of two-sided tests.

## 3. Results

### 3.1. Clinical Characteristics

The clinical characteristics of 123 patients (20 females and 103 males) are listed in Table 1. All patients with lung cancer were verified via pathology. Of the diagnosed tumors, 37 (30.08%) were adenocarcinoma, 52 (42.28%) were squamous cell carcinoma, and 34 (27.64%) were of an unknown kind. The test determined 25 (20.33%) patients who had EGFR mutations. The overall median PFS was 8.7 months for all patients. PFSs were divided into long PFS (>8.7 m) and short PFS (≤8.7 m). In the long PFS group, there were 9 females and 43 males, 50 patients with KPS80 or higher and ECOG performance 0–1, 18 patients with adenocarcinoma, 21 patients with squamous cell carcinoma, 39 patients with primary lung tumor, 18 patients with T1, and 23 patients with Stages I–II. There was no discernible difference in the features of long and short PFS (Table 2). 

### 3.2. Extraction and Selection of Radiomic Features

A total of 2042 radiomic characteristics were extracted from the radiomic dataset: 28 shape features, 415 first-order statistical features, 532 GLCM features, 330 GLDM features, 376 GLRLM features, and 361 GLSZM features. This study obtained 5 nonzero coefficient radiomic features via LASSO regression analysis. The radiomic-feature selection process of the LASSO method is shown in Figure 3. These features were PET_log_sigma_10.0.mm.3D_glrlm_ShortRunEmphasis, PET_wavelet.HLL_glszm_GrayLevelVariance, CT_log_sigma.5.0.mm.3D_glszm_ZonePercentage, CT_wavelet.LHL_g-ldm_GrayLevelVariance, CT_wavelet.LLL_firstorder_Maximum.

### 3.3. Model Performance

Logistic regression analysis shows that the T stage and overall stage (TNM) were independent predictors of PFS stratification (*p* < 0.05). The clinical EGFR model was constructed by combining the two clinical features with EGFR. The radiogenomic model was established by combining clinical EGFR and radiomic features. The results of the radiomics are visualized with a nomogram (Figure 4). The AUCs of the radiomic and radiogenomic models were 0.84 and 0.86, respectively (DeLong, *p* > 0.05). The AUC of the clinical model was 0.67 (Figure 5). The radiogenomic model was significantly superior to the clinical EGFR model (DeLong, *p* < 0.05). The ^18^F-FDG PET/CT radiogenomic model showed accuracy (ACC), sensitivity (SEN), and specificity (SPE) values, positive predictive value (PPV), and negative predictive value (NPV) of 0.78, 0.71, 0.83, 0.76, and 0.80 for discriminating PFS stratification, respectively. The radiomic model showed ACC, SEN, and SPE of 0.77, 0.71 and 0.82, PPV, and NPV of 0.74 and 0.79 for discriminating PFS stratification, respectively. For the clinical EGFR model, these were 0.65, 0.52, 0.75, 0.60, and 0.68, respectively (Table 3). Good calibration was observed for the probability of PFS stratification in the radiogenomic model (Figure 6). The decision curves (DCAs) and clinical impact curve (CIC) showed that the radiogenomic model had high clinical application value (Figure 7 and Figure 8).

### 3.4. Bootstrap Validation

The bootstrap method was used to validate the radiogenomic model. In predicting PFS stratification after SBRT in lung-cancer patients, the mean AUC of 500 internal bootstrap validations was 0.850 (95% CI, 0.849–0.851). The results show that the radiogenomic model had high accuracy.

## 4. Discussion

SBRT is an effective local-treatment method for lung cancer with the characteristics of a larger single dose and fewer fractions. Compared with long-term survival studies, the best index to evaluate the short-term efficacy of SBRT is PFS, the proportion of patients required for PFS is small, and the follow-up time is short and not affected by crossover treatment and subsequent treatment. Therefore, we chose PFS as the study endpoint. Studies on esophageal cancer [29], NSCLC [30,31], breast cancer [32], pancreatic cancer [33], nasopharyngeal carcinoma [34], and rectal cancer [35] showed the prospect of radiomics in successfully predicting the treatment prognosis of patients by extracting features from scanning images. ^18^F-FDG PET/CT imaging is essential for diagnosing and treating cancer. Numerous studies have shown that the PET/CT radiomic model could accurately predict the prognosis and best treatment of lung cancer. This suggests a future role for computer-aided diagnosis and management in oncology [36]. Lee et al. [37] used ^18^F-FDG PET/CT radiomics to perform multiblock integration with discriminant analysis, which could predict circulating tumor cells before and after SBRT for early-stage nonsmall-cell lung cancer. This is helpful in supplementing and guiding the subsequent management of early-stage NSCLC patients. A study by Dissaux et al. [38] showed that two radiomic features of ^18^F-FDG PET were independently associated with the local control of SBRT in patients with NSCLC. In PET imaging, metabolic active tumor volume (MATV), SUV, and intralesional uptake heterogeneity are quantitative indicators to evaluate glucose metabolism in malignant tumors [39]. In addition to increased metabolic activity, higher FDG uptake has been linked to various physiological characteristics, including cell proliferation, perfusion, invasiveness, and hypoxia. Therefore, radiomics could noninvasively extract several features from PET images.

Radiomic features mainly comprise four groups: (1) Intensity-statistics features that are composed of 19 features that describe the distribution of voxel intensities within the ROI using standard and simple metrics. (2) Shape features that comprise 2D and 3D features and are used to show the shape and size of the ROI. (3) Texture features that are composed of 59 features calculated with GLCM, GLDM, GLRLM, and GLSZM, and quantify the heterogeneous differences of ROI. (4) Filter and wavelet features that include intensity and texture features derived from the filter transformation and wavelet transformation of the original image, and obtained by applying filters such as wavelets. In this study, the LASSO algorithm was used to reduce the dimension and effectively avoid the overfitting phenomenon. We screened out five predictive features with great value. The radiomic features were PET_log_sigma_10.0.mm.3D_glrlm_ShortRunEmphasis, PET_wavelet.HLL_glszm_GrayLevelVariance, CT_log_sigma.5.0.mm.3D_glszm_ZonePercentage, CT_wavelet.LHL_gldm_GrayLevelVariance, CT_wavelet.LLL_firstorder_Ma-ximum. PET_log_sigma_10.0 mm. The 3D_glrlm_ShortRunEmphasis and CT_log_sigma.5.0.mm.3D_glszm_ZonePercentage are texture features obtained with the Laplacian of the Gaussian. GLRLM quantifies gray-level runs and mainly reflects the directionality and roughness of the image texture, which can indirectly reflect the heterogeneity of the tumor. The short-run emphasis element measures the distribution of short-run lengths and is associated with fine textures. GLSZM quantifies the grayscale area in the image. The element represents the ratio of the number of regions to the number of voxels in the ROI that is used to measure the roughness of the texture. PET_wavelet.HLL_glszm_GrayLevelVariance and CT_wavelet.LHL_gldm_GrayLevelVariance are texture features obtained via wavelet transform. GLDM is based on the grayscale relationship between the central pixel or voxel and its neighborhood. Gray-level variance measures the difference of gray levels in an image. Similar to GLRLM, GLDM is characterized by large and small dependence emphases that reflect heterogeneity and homogeneity, gray-level heterogeneity, and dependence uniformity, which reflect gray-level similarity and gray-level dependence in the entire ROI. GLRLM, GLSZM, and GLDM are high-order texture features that show the spatial distribution of pixels, suggesting that higher-order texture features may be more effective than lower-order texture features in reflecting the spatial heterogeneity of lung cancer lesions, thus suggesting patient efficacy.

EGFR TKIs are the first-line treatment for advanced NSCLC with an EGFR mutation. Multiple Phase III clinical trials showed that first- or second-generation EGFR TKIs were superior regarding radiographic response rates (RRs) and PFS than platinum-based chemotherapy [40]. Osimertinib was the first third-generation EGFR TKI approved for metastatic mutant EGFR NSCLC patients with the EGFR T790M resistant mutation [41]. Osimertinib exhibited better performance in the first-line therapy of advanced EGFR mutation-positive NSCLC compared to typical EGFR-TKIs. First-generation EGFR TKIs were inferior to osimertinib in terms of PFS and OS in a recent randomized Phase III FLAURA study [42]. A real-world study showed that prompt SBRT is a viable option that significantly improves the PFS for selected EGFR-mutated nonoligometastatic NSCLC patients with a stable disease during first-line EGFR-TKIs therapy [43]. Wang et al. [44] reported that combining TKIs and thoracic SBRT resulted in better PFS than that of TKIs alone in patients with EGFR-mutated multimetastatic NSCLC, with acceptable toxicity in clinical practice. The results of a prospective, single-arm, Phase II study suggest that the early SBRT of a primary lesion after EGFR-TKI therapy is a potentially safe and effective new approach for treating EGFR-mutant advanced NSCLC [45]. However, Tang et al. [46] found that, compared with patients treated with SBRT alone, patients treated with EGFR-TKIs combined with SBRT were more likely to develop radiation pneumonitis. The safety and efficacy of SBRT combined with targeted therapy must be further studied.

The prognosis for patients with the same cancer stage and the same treatment might vary greatly because individuals’ responses to treatment vary widely [47]. Predicting whether a patient would respond well to therapy early on is crucial for selecting the right interventions for patients. In this study, we established a radiogenomic model on the basis of ^18^F-FDG PET/CT radiomics and clinical EGFR to predict the PFS stratification of lung-cancer patients after SBRT treatment. The AUC of the radiogenomic model was 0.86, which was better than those of the radiomic and clinical EGFR models, which were0.84 (DeLong *p* > 0.05) and 0.67 (DeLong, *p* < 0.05), respectively. The DeLong test showed no significant difference in AUC between the radiomic and radiogenomic models, which was related to the few EGFR mutation patients in the population. The radiogenomic model’s ACC, SEN, and SPE had the same superiority. These may be beneficial in screening out patients with long PFS for further treatment. The radiogenomic model may help individuals with EGFR mutation in determining the future therapy route. However, radiogenomics in patients with lung cancer is still in its early phases, and substantial data investigations are required to confirm the hypothesis.

Several studies have established models to predict the efficacy of SBRT for lung cancer that are mainly based on clinical features or conventional imaging features. Our study proposed a new predictive model to evaluate the prognosis of SBRT in lung-cancer patients to better guide clinical decisions before SBRT treatment. The primary contribution of this study is introducing a new radiogenomic ^18^F-FDG PET/CT model (integration of radiomic characteristics and relevant clinical EGFR parameters). In this study, the radiogenomic model’s diagnostic performance and clinical benefit were better than those of the other two models, suggesting that the radiogenomic ^18^F-FDG PET/CT model may become another effective tool for predicting SBRT responses. Despite the small sample in this study, the AUC of the radiomic model was not low, most likely because many radiomic features were extracted and analyzed. Only few features were retrieved in most reported radiomic studies. To further enhance PFS stratification prediction, various candidate features could be used to extract as much information as possible from an image. These radiomic features provide the basis for LASSO to screen out highly correlated features.

This study also has some limitations. First, the study’s sample size was limited, as only 123 lung-cancer patients had undergone ^18^F-FDG PET/CT within 1 month before SBRT treatment, and only 25 had EGFR mutations. The main reason is that the high cost prevented the popularization of genetic testing and ^18^F-FDG PET/CT examination. Time constraints were also a factor. Second, due to the limited sample, to avoid overfitting, we did not group the training and validation sets. The bootstrap method was used to perform 500 resampling iterations to validate the radiogenomic model, and the mean AUC was calculated to evaluate the diagnostic performance of the model. This validation approach was used in other radiomic studies [14,48]. Third, as single-center and retrospective research, the patients recruited in this study may have been subject to selection bias and not have represented a perfectly standardized population. Fourth, segmentation is an essential step in radiomic research. This study used the manual segmentation of ^18^F-FDG PET/CT images, which may have reduced the reproducibility of this study. Although manual segmentation has high accuracy, it is time-consuming and laborious. We need a fully automatic segmentation method with high repeatability, reliability, and accuracy to solve this problem. Fifth, the sample of this study was small, and there are many risk factors related to the prognosis of lung-cancer patients, which impacts radiomic feature selection and the model construction process. Therefore, prospective, multicenter, and large-sample clinical studies are urgently needed to further clarify the clinical utility of the radiogenomic model.

## 5. Conclusions

A radiogenomic model based on ^18^F-FDG PET/CT radiomics and clinical EGFR has good application value in predicting PFS stratification of lung-cancer patients after SBRT treatment, which can provide a reference to clinical medicine.

## Figures and Tables

**Figure 1 life-13-00884-f001:**
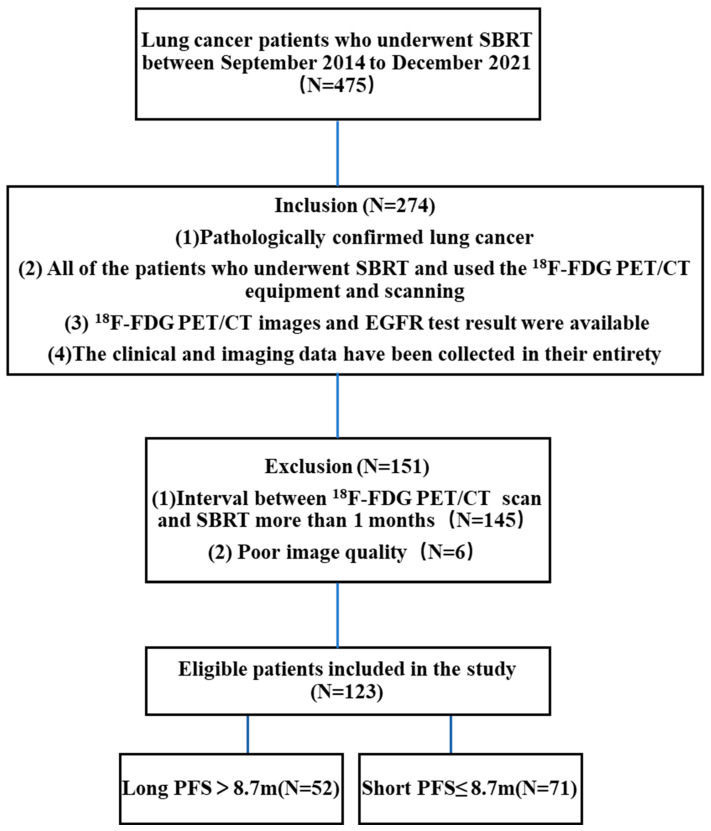
Inclusion and exclusion criteria.

**Figure 2 life-13-00884-f002:**
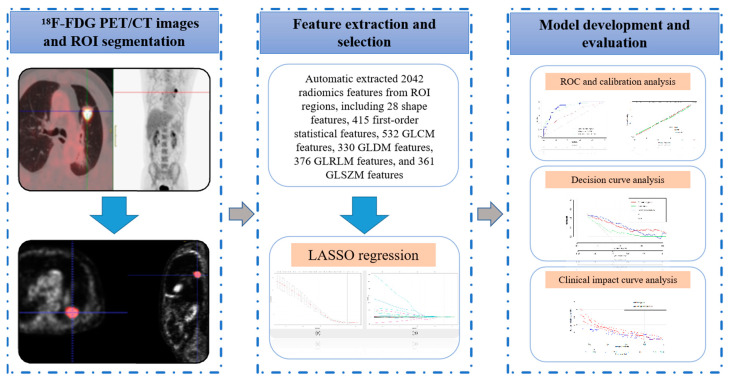
Radiomic analysis workflow.

**Figure 3 life-13-00884-f003:**
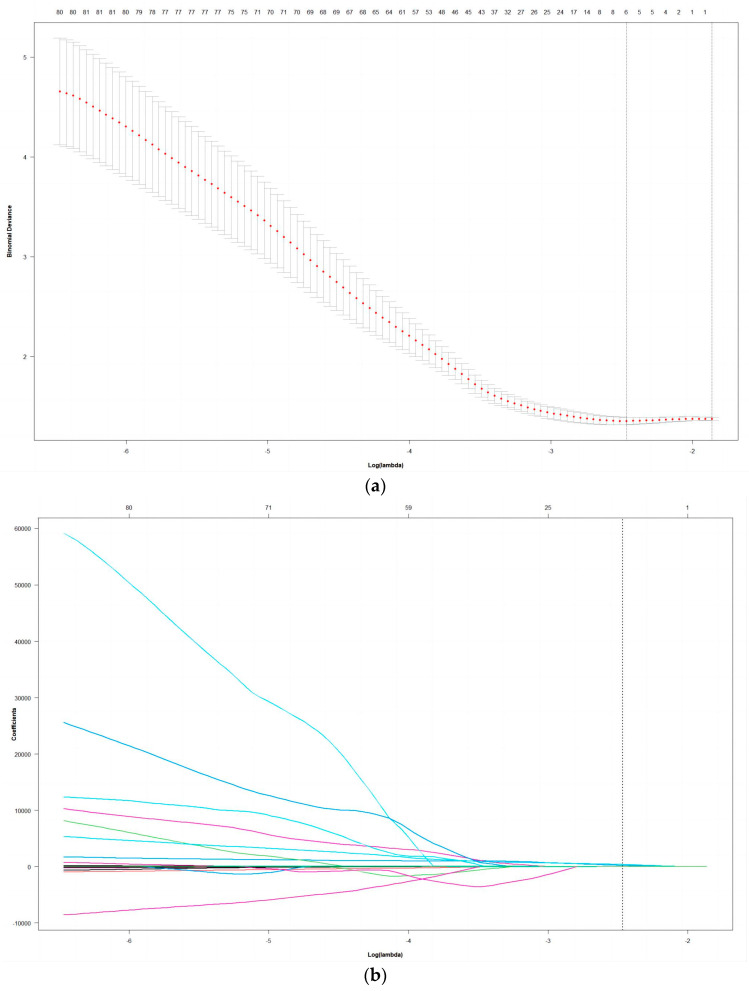
Radiomic feature selection via LASSO regression. (**a**) LASSO regression analysis of the cross-validation curves. The lower horizontal axis is log λ, the upper horizontal axis is the corresponding number of variables at different λ and the deviation value of the vertical axis; dotted vertical lines are plotted at the minimal criteria and the 1 standard error criteria; (**b**) LASSO coefficient profiles of the radiomic features. The lower abscissa is the normalized coefficient vector, and the upper abscissa is the number of nonzero coefficients in the model at this time, resulting in five features with nonzero coefficients. The ordinate is the value of the coefficient.

**Figure 4 life-13-00884-f004:**
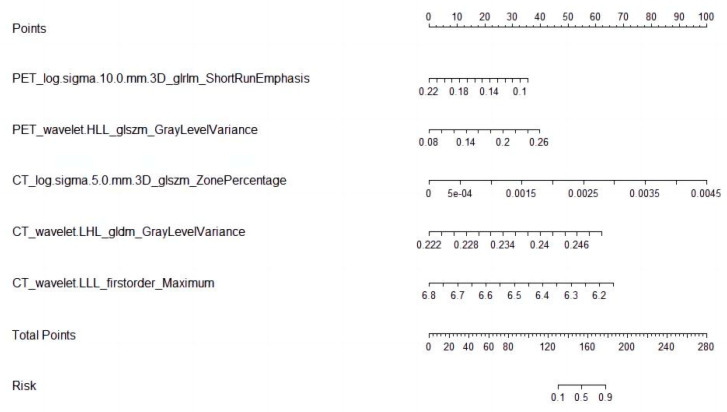
Radiomic nomogram.

**Figure 5 life-13-00884-f005:**
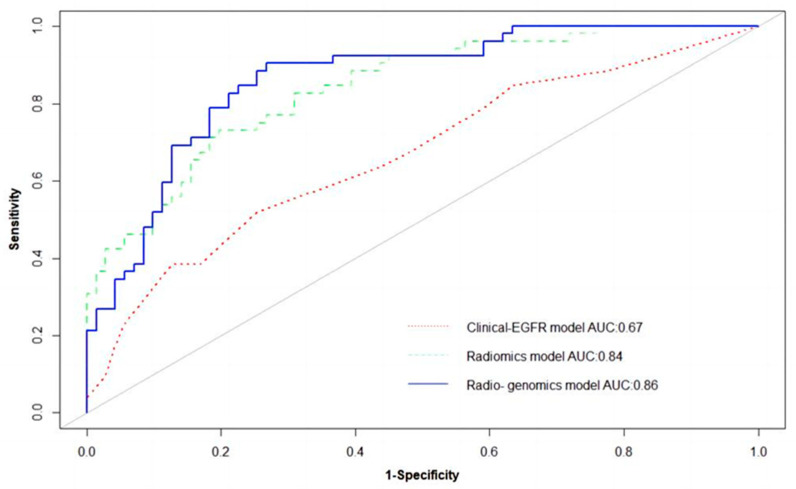
ROC of the radiomic, clinical EGFR, and radiogenomic models.

**Figure 6 life-13-00884-f006:**
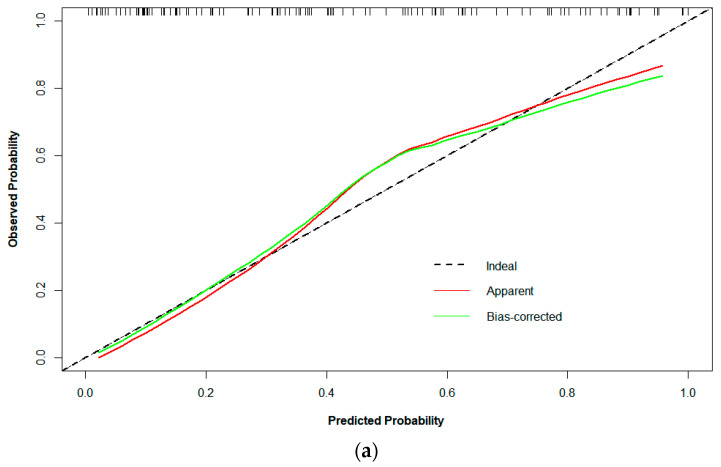
Calibration curve of the (**a**) radiomic, (**b**) clinical EGFR, and (**c**) radiogenomic models. The X axis represents the predicted PFS stratification, the y axis represents the observed PFS stratification, and the diagonal dotted line represents a perfect prediction by an ideal model. The green solid line represents the model’s performance, with a closer fit to the diagonal dotted line representing a better prediction.

**Figure 7 life-13-00884-f007:**
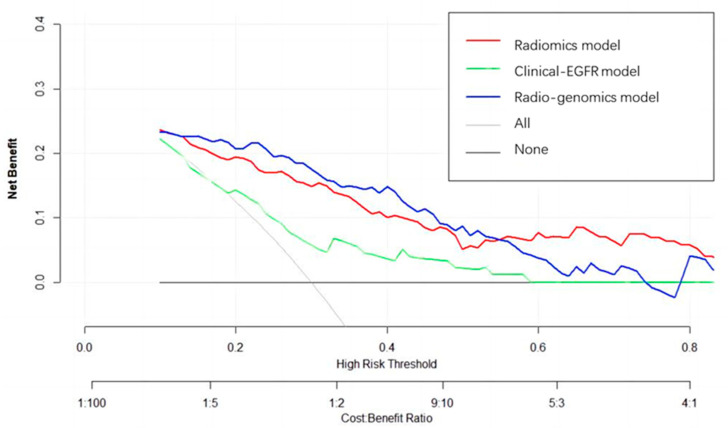
Decision curve analysis for the radiomic, clinical EGFR, and radiogenomic models. The Y axis represents the net benefit, the red line represents the radiomics model, the blue line represents the radiogenomic model, the green line represents the clinical EGFR model, the gray line represents the assumption that all patients had long PFS, and the thin black line represents the assumption that no patients had long PFS.

**Figure 8 life-13-00884-f008:**
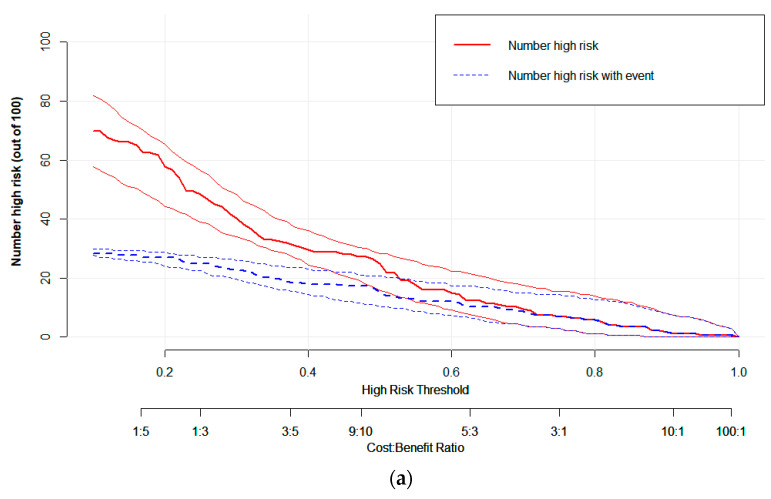
CIC visually demonstrating the estimated number of high-risk patients predicted by the (**a**) radiomic, (**b**) clinical EGFR, and (**c**) radiogenomic models, and the actual numbers for each risk threshold. The red curve shows the predicted high-risk number at different threshold probabilities, and the blue curve represents actual high-risk patients.

**Table 1 life-13-00884-t001:** Clinical characters of lung-cancer patients (n = 123).

Characteristics	Numbers of Cases (n)	Constituent Ratio (%)
Age, years (≤70 y/>70 y)	63/60	51.22/48.78
Sex (male/female)	103/20	83.74/16.26
KPS (100/90/80/70)	2/40/78/3	1.63/32.52/63.41/2.44
ECOG performance status (0/1/2)	47/74/2	38.21/60.16/1.63
Cigarette smoking (ever been smoker/never smoker)	40/83	32.52/67.48
Tumor location (central/peripheral)	62/61	50.41/49.59
Pathological type (adenocarcinoma/squamous cell carcinoma/unknown)	37/52/34	30.08/42.28/27.64
Source of tumor (primary/metastatic)	94/29	76.42/23.58
EGFR mutation (positive/negative)	25/98	20.33/79.67
T classification (T1–T4)	51/32/22/18	41.46/26.02/17.89/14.63
Overall stage (I–II/III–IV)	43/80	34.96/65.04
Maximal SUV (≤9.6/>9.6)	63/60	51.22/48.78
BED (≤100/>100)	74/49	60.16/39.84

**Table 2 life-13-00884-t002:** Characteristics of patients with different PFS stratification.

Characteristics	Long PFS (n = 52)	Short PFS (n = 71)	*p*
Age (≤70 y/>70 y)	24/28	39/32	0.34
Sex (female/male)	9/43	11/60	0.81
KPS (100/90/80/70)	1/20/29/2	1/20/49/1	0.43
ECOG performance status (0/1/2)	22/28/2	25/46/0	0.15
Cigarette smoking (ever/never)	36/16	47/24	0.85
Tumor location (central/peripheral)	29/23	55/16	0.36
Pathology (adenocarcinoma/squamous cell carcinoma/unknown)	18/21/13	19/31/21	0.68
Source of tumor (primary/metastatic)	39/13	55/16	0.83
T classification (1–4)	18/14/10/10	33/18/12/8	0.49
Overall stage (I–II/III–IV)	23/29	20/51	0.07
Maximal SUV (≤9.6/>9.6)	28/24	35/36	0.62
BED (≤100/>100)	36/16	38/33	0.08

**Table 3 life-13-00884-t003:** The performance of the radiomic, clinical EGFR, and radiogenomic models.

Models	AUC	ACC	SEN	SPE	PPV	NPV	Delong’s P
Radiogenomic model	0.86	0.78	0.71	0.83	0.76	0.80	
Radiomic model	0.84	0.77	0.71	0.82	0.74	0.79	>0.05
Clinical EGFR model	0.67	0.65	0.52	0.75	0.60	0.68	<0.05

## Data Availability

The original contributions presented in the study are included in the article. Further inquiries can be directed to the corresponding authors.

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
