# Peer review of "Predicting the Efficacy of SBRT for Lung Cancer with 18F-FDG PET/CT Radiogenomics"

_life, 2023, doi:10.3390/life13040884_

Round 1

Reviewer 1 Report

The presented subject is important for clinical practise, however, several issues need clarification:

“This retrospective study received approval from the institution's ethical committee, 75 and informed patient consent was not necessary”. – it has to be clarified;

“Exclusion criteria: (1) poor 18F-FDG PET/CT picture quality” – should be defined more precisely;

Diagram showing the inclusion and exclusion criteria. – numbers of excluded patients should be written;

Fig. 3 – complete the description a) and b);

Editorial correction is necessary;

All tables and figures in Results section should be described in more detail.

Discussion related to the best radiomics features in this study should be extended.

How exactly was the validation of the model done?

Author Response

1.The presented subject is important for clinical practice, however, several issues need clarification: “This retrospective study received approval from the institution's ethical committee, 75 and informed patient consent was not necessary”. – it has to be clarified;

Response: Special thanks to the reviewer have contributed their time and talent to the creation and review of this article. Thank you to our affirmation. The statements have been corrected. The supplementary content is as follows: The need for informed consent was waived due to retrospective collection of data. All data on patients were provided by Zhejiang Taizhou Hospital. (Line 459-461)

2.“Exclusion criteria: (1) poor 18F-FDG PET/CT picture quality” – should be defined more precisely;

Response: Special thanks to the reviewer have contributed their time and talent to the creation and review of this article. Thank you for your sincere suggestion. We have modified the manuscript, hoping to help the reviewer understand better. The changes are as follows: poor 18F-FDG PET/CT picture quality (prominent artifacts) makes texture analysis and tumor region of interest (ROI) delineation problematic, preventing imaging radiomics analysis. (Line 81-83)

3.Diagram showing the inclusion and exclusion criteria. – numbers of excluded patients should be written;

Response: Special thanks to the reviewer have contributed their time and talent to the creation and review of this article. As same as considered by the reviewer, such thoughts have been entertained by us. We have modified the Figure 1, hoping to help the reviewer understand better. Thank you again for the reviewer’s kind guidance. (Line 90)

4.Fig. 3 – complete the description a) and b);

Response: We sincerely thank the reviewer for careful reading. We agree with the reviewer's statement. The changes are as follows: (Line 230-237) Radiomics feature selection by LASSO regression. a) LASSO regression analysis of the cross-validation curves. The lower horizontal axis is log λ, the upper horizontal axis is the corresponding number of variables at different λ, and the deviation value of the vertical axis, dotted vertical lines are plotted at the minimum criteria and the 1- standard error criteria; b) LASSO coefficient profiles of the radiomics features. The lower abscissa is the normalized coefficient vector, and the upper abscissa is the number of non-zero coefficients in the model at this time, resulting in 5 features with non-zero coefficients. The ordinate is the value of the coefficient.

5.Editorial correction is necessary;

Response: We worked on the manuscript for long time and the repeated addition and removal of sentences and sections obviously led to poor readability and have now worked on both language and readability. We really hope that the flow and language level have been substantially improved.

6.All tables and figures in Results section should be described in more detail.

Response: We sincerely thank the reviewer for careful reading. We agree with the reviewer's statement. The supplementary contents are as follows:(Line 211-215) In the long PFS group, there were 9 females and 43 males, 50 patients with KPS80 or higher and ECOG performance 0-1, 18 patients with adenocarcinoma, 21 patients with squamous cell carcinoma, 39 patients with primary lung tumor, 18 patients with T1, and 23 patients with stage I-II.(Line 223-225) This study obtained 5 non-zero coefficient radiomics features by LASSO regression analysis. The radiomics features selection process of the LASSO method is shown in Figure 3.

7.Discussion related to the best radiomics features in this study should be extended.

Response: Special thanks to the reviewer have contributed their time and talent to the creation and review of this article. Thank you for your sincere suggestion. We accept it humbly. The supplementary contents are as follows: (Line 325-350) In this study, the LASSO algorithm was used to reduce the dimension and effectively avoid the overfitting phenomenon. 5 predictive features with great value were screened out. Five non-zero coefficient radiomics features were obtained after radiomics analysis of 18F-FDG PET/CT images of lung cancer patients undergoing SBRT in this study. The radiomics features were PET_log_sigma_10.0.mm.3D_glrlm_ShortRunEmphasis,  PET_wavelet.HLL_glszm_GrayLevelVariance,CT_log_sigma.5.0.mm.3D_glszm_ZonePercentage,CT_wavelet.LHL_gldm_GrayLevelVariance,CT_wavelet.LLL_firstorder_Ma-ximum.PET_log_sigma_10.0mm.3D_glrlm_ShortRunEmphasis and CT_log_sigma.5.0.mm.3D_glszm_ZonePercentage are texture features obtained by the laplacian of gaussian. GLRLM quantifies gray level runs, and mainly reflects the directionality and roughness of the image texture, which can indirectly reflect the heterogeneity of the tumor. The element Short Run Emphasis measures the distribution of short run lengths, and is associated with fine textures. GLSZM quantifies the grayscale area in the image. The element represents the ratio of the number of regions to the number of voxels in the ROI, which is used to measure the roughness of the texture. PET_wavelet.HLL_glszm_GrayLevelVariance and CT_wavelet.LHL_gldm_GrayLevelVariance are texture features obtained by wavelet transform. GLDM is based on the grayscale relationship between the central pixel or voxel and its neighborhood. Gray Level Variance measures the difference of gray levels in an image. Similar to GLRLM, GLDM is characterized by large dependence emphasis and small dependence emphasis that reflect heterogeneity and homogeneity, gray-level heterogeneity, and dependence uniformity that reflect gray-level similarity and gray-level dependence in the entire ROI. GLRLM, GLSZM and GLDM are both high-order texture features that show the spatial distribution of pixels, suggesting that high-er-order texture features may be more effective than lower-order texture features in reflecting the spatial heterogeneity of lung cancer lesions, thus suggesting patients’ efficacy.

8.How exactly was the validation of the model done?

Response: We sincerely thank the reviewer for careful reading. We have modified the manuscript, hoping to help the reviewer understand better. (Line 189-196)

The bootstrap method for internal validation is to construct a bootstrap resampling sample with the same sample size in the model development cohort. This sample is used as the validation set. The model development cohort was used as the training set to evaluate the model performance and repeat this process n times to obtain the performance of the model in internal validation. Based on theoretical basis and literature support, it is the best way to choose bootstrap method for internal validation to evaluate the stability of the model under the condition of insufficient sample size [1-3]. Therefore, we did not set the training set and the validation set in the conventional way, but chose rely on the bootstrap method.

Peroration: We tried our best to improve the manuscript and made some changes marked in red in revised paper. We appreciate for Reviewers’ warm work earnestly, and hope the correction will meet with approval. Once again, thank you very much for your comments and suggestion.

Reference

  1. Tomicic, A.; Malesevic, A.; Cartolovni, A. Ethical, Legal and Social Issues of Digital Phenotyping as a Future Solution for Present-Day Challenges: A Scoping Review. Sci Eng Ethics 2021, 28 (1), 1.doi:10.1007/s11948-021-00354-1.
  2. Chen, Y. H.; Wang, T. F.;  Chu, S. C.;  Lin, C. B.;  Wang, L. Y.;  Lue, K. H.;  Liu, S. H.; Chan, S. C. Incorporating radiomic feature of pretreatment 18F-FDG PET improves survival stratification in patients with EGFR-mutated lung adenocarcinoma. PLoS One 2020, 15 (12), e0244502.doi:10.1371/journal.pone.0244502.
  3. Wang, G.; He, L.;  Yuan, C.;  Huang, Y.;  Liu, Z.; Liang, C. Pretreatment MR imaging radiomics signatures for response prediction to induction chemotherapy in patients with nasopharyngeal carcinoma. Eur J Radiol 2018, 98, 100-106.doi:10.1016/j.ejrad.2017.11.007.

Reviewer 2 Report

# General Comments

Thank you for the opportunity to review this manuscript. The authors reported on predicting efficacy oof SBRT for lung cancer by 18F-FDG PET/CT radio-genomics. I think that this report requires some revisions to describe this study more appropriately.

# Major points

1.       I think one of the limitations of this study is that the patients with short follow-up period are included, though the authors set the endpoint as PFS.

2.       As for the feature selection algorithms, why the authors choose LASSO?

3.       How did the authors determine the number of radiomic features to build the model?

4.       The authors compared the characteristics of the patients with long PFS and short PFS. The authors are required to describe the definition of PFS in the Materials and Methods section. Furthermore, I wondered why the authors did not used log-rank test for survival comparison.

5.       How the authors split the training data and the validation data? Please describe this.

6.       Did the authors try any other types of feature selection algorithms and models considering survival time?

Author Response

1.I think one of the limitations of this study is that the patients with short follow-up period are included, though the authors set the endpoint as PFS.

Response: Thank you for your careful reading. SBRT as a local treatment modality, the best index to evaluate the short-term efficacy is PFS. Compared with survival studies, the proportion of patients required for PFS is small and the follow-up time is short, which is not affected by crossover treatment and subsequent treatment. Therefore, we chose PFS as the study endpoint in this study. (Line 291-296)

2.As for the feature selection algorithms, why the authors choose LASSO?

Response: We sincerely thank the reviewer for careful reading. LASSO, compressing the coefficients of features and making some regression coefficients turn to 0 to achieve the purpose of feature selection, which can be widely used in model improvement and selection. LASSO can remove invalid variables (the coefficient is zero, that is invalid), which can not only solve the problem of overfitting, but also directly reduce some repeated unnecessary parameters to zero in the process of parameter reduction, so as to achieve the role of extracting useful features. Compared with the traditional methods, LASSO algorithm performs better in variable selection under the condition that the number of training samples is small, there are a large number of similar features in the sample features, and the sum of absolute values of model coefficients is less than a constant. In addition, the method of LASSO is more conventional, and it can be seen in most articles that LASSO is used for feature selection at present[1-3]. Therefore, we chose LASSO as the way of feature selection.

3.How did the authors determine the number of radiomic features to build the model?

Response: Special thanks to the reviewer have contributed their time and talent to the creation and review of this article.

Firstly, the model is a combination of different features, so the more features, the more parameters of the model, and then the complexity of the model will increase. In the case of limited sample size, fitting a model with many parameters will cause the model to learn not only the data distribution, but also the noise distribution, so overfitting will occur. Therefore, based on theory and previous practice, LASSO was selected as the feature selection method [1-3].

Then, to improve the performance of the model, a tenfold cross-validation of the model was carried out in the study, the best λ was obtained during the cross-validation procedure. The tuning parameter (λ) in the LASSO model was selected using 10-fold cross validation through the minimum criterion. The relationship between the area under the receiver operating characteristic (AUC) curve and log (λ) was plotted. Vertical dashed lines were drawn at the best values using the minimum standard and 1 standard deviation of the minimum standard (as showed in Figure 3a). Coefficient profiles were plotted against the log (λ) series. Vertical lines were drawn at selected values using 10-fold cross-validation, where the best λ yielded nonzero coefficients (as showed in Figure 3b). These variables were used as the main influencing factors of radiomics score (Rad-score). (Line 230-237)

4.The authors compared the characteristics of the patients with long PFS and short PFS. The authors are required to describe the definition of PFS in the Materials and Methods section. Furthermore, I wondered why the authors did not used log-rank test for survival comparison.

Response: We sincerely thank the reviewer for careful reading.

As showed in manuscript at line 87-89, “ Time until disease progression (e.g., growth of a residual tumor or formation of a new metastatic lesion) or death or censoring at the date of the final follow-up, after SBRT was initiated was used to calculate PFS ”.

  SBRT as a local treatment modality, the best index to evaluate the short-term efficacy is PFS. Compared with survival studies, the proportion of patients required for PFS is small and the follow-up time is short, which is not affected by crossover treatment and subsequent treatment. Therefore, we chose PFS as the study endpoint in this study, and without perform survival comparison. (Line 291-296)

5.How the authors split the training data and the validation data? Please describe this.

Response: We sincerely thank the reviewer for careful reading. We have modified the manuscript, hoping to help the reviewer understand better. (Line 189-196)

The bootstrap method for internal validation is to construct a bootstrap resampling sample with the same sample size in the model development cohort. This sample is used as the validation set. The model development cohort was used as the training set to evaluate the model performance and repeat this process n times to obtain the performance of the model in internal validation. Based on theoretical basis and literature support, it is the best way to choose bootstrap method for internal validation to evaluate the stability of the model under the condition of insufficient sample size [4-6]. Therefore, we did not set the training set and the validation set in the conventional way, but chose rely on the bootstrap method.

6.Did the authors try any other types of feature selection algorithms and models considering survival time?

Response: Special thanks to the reviewer have contributed their time and talent to the creation and review of this article. Thank you for your sincere suggestion.

SBRT as a local treatment modality, the best index to evaluate the short-term efficacy is PFS. Compared with survival studies, the proportion of patients required for PFS is small and the follow-up time is short, which is not affected by crossover treatment and subsequent treatment. In addition, this study is a retrospective study, and the subsequent intervention of patients affects the survival of patients, which also affects the judgment of the efficacy of SBRT, which is different from the purpose of our study. Therefore, we chose PFS as the study endpoint in this study. In the future, we can further attempt in prospective, multi-center, large-sample data sets.

Peroration: We tried our best to improve the manuscript and made some changes marked in red in revised paper. We appreciate for Reviewers’ warm work earnestly, and hope the correction will meet with approval. Once again, thank you very much for your comments and suggestion.

Reference

  1. Peng, G.; Zhan, Y.;  Wu, Y.;  Zeng, C.;  Wang, S.;  Guo, L.;  Liu, W.;  Luo, L.;  Wang, R.;  Huang, K.;  Huang, B.;  Chen, J.; Chen, C. Radiomics models based on CT at different phases predicting lymph node metastasis of esophageal squamous cell carcinoma (GASTO-1089). Front Oncol 2022, 12, 988859.doi:10.3389/fonc.2022.988859.
  2. Lin, M.; Tang, X.;  Cao, L.;  Liao, Y.;  Zhang, Y.; Zhou, J. Using ultrasound radiomics analysis to diagnose cervical lymph node metastasis in patients with nasopharyngeal carcinoma. Eur Radiol 2023, 33 (2), 774-783.doi:10.1007/s00330-022-09122-6.
  3. Tikhonova, V. S.; Karmazanovsky, G. G.;  Kondratyev, E. V.;  Gruzdev, I. S.;  Mikhaylyuk, K. A.;  Sinelnikov, M. Y.; Revishvili, A. S. Radiomics model-based algorithm for preoperative prediction of pancreatic ductal adenocarcinoma grade. Eur Radiol 2023, 33 (2), 1152-1161.doi:10.1007/s00330-022-09046-1.
  4. Tomicic, A.; Malesevic, A.; Cartolovni, A. Ethical, Legal and Social Issues of Digital Phenotyping as a Future Solution for Present-Day Challenges: A Scoping Review. Sci Eng Ethics 2021, 28 (1), 1.doi:10.1007/s11948-021-00354-1.
  5. Chen, Y. H.; Wang, T. F.;  Chu, S. C.;  Lin, C. B.;  Wang, L. Y.;  Lue, K. H.;  Liu, S. H.; Chan, S. C. Incorporating radiomic feature of pretreatment 18F-FDG PET improves survival stratification in patients with EGFR-mutated lung adenocarcinoma. PLoS One 2020, 15 (12), e0244502.doi:10.1371/journal.pone.0244502.
  6. Wang, G.; He, L.;  Yuan, C.;  Huang, Y.;  Liu, Z.; Liang, C. Pretreatment MR imaging radiomics signatures for response prediction to induction chemotherapy in patients with nasopharyngeal carcinoma. Eur J Radiol 2018, 98, 100-106.doi:10.1016/j.ejrad.2017.11.007.

Round 2

Reviewer 1 Report

Editorial check required. I am satisfied with methodological corrections.

Reviewer 2 Report

# General Comments

Thank you for the opportunity to review the revised manuscript. The authors have responded appropriately to our comments and this manuscript is currently suitable for publication.